# Knowledge, Attitudes, and Acceptance of COVID-19 Vaccines among Secondary School Pupils in Zambia: Implications for Future Educational and Sensitisation Programmes

**DOI:** 10.3390/vaccines10122141

**Published:** 2022-12-14

**Authors:** Steward Mudenda, Moses Mukosha, Brian Godman, Joseph O. Fadare, Olayinka O. Ogunleye, Johanna C. Meyer, Phumzile Skosana, Jacob Chama, Victor Daka, Scott K. Matafwali, Billy Chabalenge, Bwalya A. Witika

**Affiliations:** 1Department of Pharmacy, School of Health Sciences, University of Zambia, Lusaka P.O. Box 50110, Zambia; 2Department of Public Health Pharmacy and Management, School of Pharmacy, Sefako Makgatho Health Sciences University, Pretoria 0208, South Africa; 3Centre of Medical and Bio-Allied Health Sciences Research, Ajman University, Ajman 346, United Arab Emirates; 4Department of Pharmacoepidemiology, Strathclyde Institute of Pharmacy and Biomedical Science (SIPBS), University of Strathclyde, Glasgow G4 0RE, UK; 5Department of Pharmacology and Therapeutics, Ekiti State University College of Medicine, Ado-Ekiti 362103, Nigeria; 6Department of Medicine, Ekiti State University Teaching Hospital, Ado-Ekiti 362103, Nigeria; 7Department of Pharmacology, Therapeutics and Toxicology, Lagos State University College of Medicine, Lagos 100271, Nigeria; 8Department of Medicine, Lagos State University Teaching Hospital, Lagos 100271, Nigeria; 9South African Vaccination and Immunisation and Centre, Sefako Makgatho Health Sciences University, Pretoria 0208, South Africa; 10Department of Clinical Pharmacy, School of Pharmacy, Sefako Makgatho Health Sciences University, Pretoria 0208, South Africa; 11Department of Public Health, Michael Chilufya Sata School of Medicine, Copperbelt University, Ndola P.O. Box 71191, Zambia; 12Clinical Research Department, Faculty of Infectious and Tropical Diseases, London School of Hygiene & Tropical Medicine, Keppel Street, London WC1E 7HT, UK; 13Department of Medicines Control, Zambia Medicines Regulatory Authority, Lusaka P.O. Box 31890, Zambia; 14Department of Pharmaceutical Sciences, School of Pharmacy, Sefako Makgatho Health Sciences University, Pretoria 0208, South Africa

**Keywords:** adolescents, attitude, children, COVID-19 vaccines, hesitancy, knowledge, pupils, secondary schools, vaccine acceptance, Zambia

## Abstract

The coronavirus disease 2019 (COVID-19) pandemic resulted in the closure of schools to slow the spread of the virus across populations, and the administration of vaccines to protect people from severe disease, including school children and adolescents. In Zambia, there is currently little information on the acceptance of COVID-19 vaccines among school-going children and adolescents despite their inclusion in the vaccination programme. This study assessed the knowledge, attitudes, and acceptance of COVID-19 vaccines among secondary school pupils in Lusaka, Zambia. A cross-sectional study was conducted from August 2022 to October 2022. Of the 998 participants, 646 (64.7%) were female, and 127 (12.7%) would accept to be vaccinated. Those who were willing to be vaccinated had better knowledge (68.5% vs. 56.3%) and a positive attitude (79.1% vs. 33.7%) compared to those who were hesitant. Overall, the odds of vaccine acceptance were higher among pupils who had higher knowledge scores (AOR = 11.75, 95% CI: 6.51–21.2), positive attitude scores (AOR = 9.85, 95% CI: 4.35–22.2), and those who knew a friend or relative who had died from COVID-19 (AOR = 3.27, 95% CI: 2.14–5.09). The low vaccine acceptance among pupils is of public health concern, emphasising the need for heightened sensitisation programmes that promote vaccine acceptance among pupils in Zambia.

## 1. Introduction

The coronavirus disease 2019 (COVID-19) pandemic appreciably increased morbidity and mortality as well as associated costs [1,2,3,4,5,6,7]. Public health measures, including lockdowns, border closures, contact tracing and quarantining measures as well as the wearing of personal protective equipment, were introduced across countries, including among African countries, in an attempt to slow the spread of the severe acute respiratory coronavirus type 2 (SARS-CoV-2) in the absence of effective treatments and vaccines [8,9,10,11,12,13].

Vaccines were developed as one of the key solutions to prevent severe disease, hospitalisation, and death due to COVID-19 along with controlling the spread of the disease in the absence of effective treatments [14,15,16,17,18,19,20,21]; however, this could be changing [22,23,24,25,26,27]. These combined developments should help to address the adverse consequences of lockdown and social distancing measures, including the closure of clinics, which in turn negatively impacted routine immunisation programmes among children, including those in Africa [28,29]. Alongside this, helping to address the negative impact on education following school closures and concerns exacerbated by the lack of computers and the costs of internet bundles across Africa [30,31]. These concerns resulted in the development, deployment and administration of COVID-19 vaccines across countries [15,32,33,34,35,36,37,38], although there have been concerns with the vaccines and their uptake in some countries [34,35,39].

Vaccines are critical in addressing vaccine-preventable disease outbreaks caused by micro-organisms [38,40,41,42]. However, the use of vaccines and their success in the immunisation of populations requires that individuals are sensitised to their importance, have good knowledge about them and have confidence in their safety and efficacy [43,44,45,46]. This though is not always the case among the general population as well as among healthcare workers (HCWs), some of whom have been hesitant to receive COVID-19 vaccines despite their undoubted effectiveness and good safety profiles [46,47,48,49,50].

Currently though, acceptance of COVID-19 vaccines is variable across countries [34,35,39,51,52,53,54], with many individuals hesitant due to a lack of awareness of the vaccines and their potential impact, limited knowledge and concerns regarding their safety and effectiveness, negative attitudes towards the vaccines as well as fears of adverse effects [35,46,55,56,57]. As a result, high hesitancy and low vaccination rates have been reported across countries and continents including Africa [39,48,58,59,60,61,62,63,64,65], exacerbated by the spreading of misinformation and myths [39,64,66,67,68,69]. In addition, the development of mutations among the Alpha, Beta, Gamma, Delta, and Omicron variants has raised further questions about the effectiveness of vaccines long-term and also contributed to hesitancy [70,71]. Consequently, this requires that all vaccine candidates that are being developed must take into consideration the evolution of SARS-CoV-2 variants [70,72].

Among school-going children and adolescents, there is currently limited information available about vaccines, including those for COVID-19 [73,74], which can affect children’s knowledge, attitudes, and acceptance of vaccines. Consequently, there is a need to develop strategies that will improve vaccine acceptance among school pupils including for COVID-19, specially tailored to hesitant children [75,76]. Potential strategies include improving secondary school pupils’ knowledge aboutCOVID-19 infections as studies have shown poor knowledge among the hesitant groups [77,78]. On the other hand, a study reported a vaccination acceptance rate of over 50% among older pupils, with only 12.9% of surveyed pupils opting out of vaccination [76]. Understanding these issues going forward is important. We do know that preventive measures associated with the COVID-19 pandemic negatively impacted secondary school pupils and educational systems generally [77,79,80], affecting nearly 1.6 billion school students globally [81]. Consequently, this needs to be urgently addressed, with effective vaccines part of the strategy to alleviate the need for future lockdown and other measures including new potential treatments [38,82,83].

Zambia, a country in sub-Saharan Africa, reported its first case of COVID-19 on 18 March 2020. Concerns with COVID-19 and its impact in Zambia led to the introduction of preventive measures, including the closure of schools as well as the implementation of other control measures [84,85]. However, the morbidity and mortality associated with COVID-19 increased significantly during the second and third waves in Zambia, which is a concern [11,85]. The rollout and administration of COVID-19 vaccines commenced in April 2021in Zambia [39,86]. Since then, vaccine acceptance rates of 33.4% and 66%have been reported across the general population in Zambia [62,86] and 24.5% among university students, respectively [63]. In January 2022, the health authorities in Zambia started vaccinating secondary school children and adolescents [39]. However, there is currently a dearth of information regarding the knowledge, attitudes, and acceptance of COVID-19 vaccines among children and adolescents attending secondary schools in Zambia. Consequently, this study aimed to address this deficit by assessing the knowledge, attitudes, and acceptance of COVID-19 vaccines among secondary school pupils in Zambia. Without this baseline information, it would be difficult to design programmes to increase vaccine acceptance and uptake among secondary school pupils where this is an issue, with hesitancy concerns already reported in East Africa among citizens not educated above secondary school levels [87]. 

## 2. Materials and Methods

### 2.1. Study Design, Setting and Population

This cross-sectional study was conducted among secondary school pupils in Lusaka, Zambia, from August 2022 to October2022, following the rollout of COVID-19 vaccination in January 2022, amongst children and adolescents aged from 12 to 17 years [39]. In Zambia, secondary schools educate pupils from Grade 8 to Grade 12.

Since Lusaka is the capital city of Zambia, it was purposefully selected for this study, assuming that should there be knowledge, attitude, and acceptance deficits concerning COVID-19 vaccines among pupils attending secondary schools in Lusaka, this is likely to be worse among rural-dwelling pupils. All vaccinated pupils (29.2%, n = 411) were excluded from this study.

### 2.2. Sampling and Sample Size Consideration

A multi-stage random sampling approach was used in this study. Firstly, we randomly selected 32 secondary schools from a total of 111 secondary schools in Lusaka city. From each school, 2–4 classes were randomly selected to participate in the research using proportion to school size. From each class, all potential pupils were considered for sampling using a simple random sampling technique to ensure that each pupil in the class will have the same chance of being selected for the study. Before conducting the study, a representative sample size was estimated using Ausvet Raosoft software (http://www.raosoft.com/samplesize.html (accessed on 22 July 2022)). The sample size was estimated at a 95% confidence level, with a margin of error of 5%, and a finite population of 10,000 for the locality. A 10% incomplete, loss, or non-response was taken into consideration. With an assumed moderate design effect of 1.5, a minimum sample size of 814 pupils was estimated. The participants had to be registered in a secondary school in Lusaka, Zambia, during the study to be eligible for the study.

### 2.3. Data Collection Tool

This study used a validated self-administered questionnaire from a similar study consisting of four parts [88]. Part I had five questions on the sociodemographic characteristics of participants; Part II had five questions regarding the knowledge of participants concerning the COVID-19 vaccines with yes or no response options; Part III had five questions on the attitudes of participants towards COVID-19 with “yes” or “no” or “I do not know” response options, and Part IV had questions on factors that affect acceptance of COVID-19 vaccines among secondary school pupils. Finally, vaccine acceptance was assessed by the question, “Would you accept to be vaccinated against COVID-19?” as shown in Appendix A.

To check for the simplicity of the questions, we conducted a pilot study among 50 pupils from different secondary schools in Lusaka. These pupils did not form part of the principal study. Each child took approximately 20 to 30 min to respond to the questionnaire. The questionnaire had a Cronbach’s alpha of 0.76 for knowledge and 0.84 for attitude scales, indicating acceptable reliability.

The questionnaire was subsequently distributed to all unvaccinated eligible pupils in the selected schools after they provided assent. The consent to participate in the study was given by the pupils’ parents and/or their guardians. Data collection was undertaken by two data collectors who were trained by the main author (SM).

### 2.4. Data Management and Analysis

Stata version 17/BE (Stata Corp., College Station, TX, USA) was used for the statistical analysis. All analyses accounted for the clustering of pupils within schools by using robust estimation of standard errors.

Knowledge and attitude scales were scored as follows; for each correct answer, a “yes” for knowledge questions and a positive “yes” for attitude questions were assigned a score of one, while an incorrect “no” for knowledge and a negative “no/don’t know” for attitude questions were assigned a score of zero. The knowledge and attitude scores were subsequently calculated as the sum of the total scores from all the questions.

Continuous variables (age, knowledge and attitude score) were summarised using means and 95% confidence intervals (95% CI) and whether or not the pupil would accept COVID-19 vaccination. We fitted logistic regression models with robust estimation of standard errors with “COVID-19 vaccine acceptance” as the outcome variable and one of the predictor variables at a time, adjusting for age, to assess for any evidence of an association between the variable and COVID-19 vaccine acceptance.

Following this, a multivariable logistic regression model was fitted with the knowledge and attitude scores, age and other variables that were significant in single age-adjusted models. Interactions between knowledge and attitude scores and the confounding variables that remained in the final model were considered one by one.

## 3. Results

The study enrolled 998 (95% response rate) unvaccinated pupils of whom 127(12.7%) would accept the COVID-19 vaccine if it was made available. The largest proportion of those who would accept the vaccine were females (85; 66.9%) in Grade 8 (39; 30.7%) and with a mean age of 15.3 years [95% CI: 15.0–15.6].

The characteristics of the surveyed pupils, their socio-demographics, sources of information about COVID-19, and average total scores of knowledge and attitudes towards the COVID-19 vaccine are provided in Table 1. Overall, pupils who would accept vaccination reported good knowledge (68.5% vs. 56.3%) and positive attitude scores (79.1% vs. 33.7%)compared to those who would refuse vaccination.

The pupils’ experiences during the COVID-19 pandemic are summarized in Table 2. The majority 855 (85.7%) of the participants had not suffered from COVID-19 but among these, 467 (46.8%) knew a friend or relative who had previously suffered from COVID-19. A small proportion of participants 158 (15.8%) reported knowing a relative or friend who had died of COVID-19, and 558 (55.9%) mentioned that preventive measures were not stressful to follow. A larger proportion 779 (78.1%) were able to practice social distancing and 710 (71.1%) were never in quarantine during the pandemic.

The logistic regression model that adjusted for age, considering one variable at a time, found that attitude and knowledge scores, knowing a friend/relative who died from COVID-19, being in quarantine due to COVID-19 infection and having a chronic condition were associated with COVID-19 vaccine acceptance (Table 3).

After controlling for the modifying variables that were statistically significant at the 5% level in the univariable logistic regression model (age, attitude and knowledge score, knowing a friend/relative who died from COVID-19, being in quarantine due to COVID-19 and having a chronic condition), the multivariable logistic regression model showed that independent factors associated with COVID-19 vaccine acceptance were knowledge, attitudes, knowing a friend or relative who died from COVID-19 and being in Grade 9 compared to Grade 8.

Pupils with higher knowledge scores (AOR = 11.75, 95% CI: 6.51–21.2), higher attitude scores (AOR = 9.85, 95% CI: 4.35–22.2) and those who knew a friend or relative who died from COVID-19 (AOR = 3.27, 95% CI: 2.14–5.09) were more likely to accept a COVID-19 vaccine. However, being in Grade 9 compared to Grade 8 (AOR = 0.45, 95% CI: 0.22–0.93) was associated with lower odds of accepting the COVID-19 vaccine.

## 4. Discussion

To the best of our knowledge, this is the first study conducted in Zambia to assess the knowledge, attitudes, and acceptance of COVID-19 vaccines among pupils attending secondary schools in Zambia. We found that only 12.7% of surveyed pupils would accept to be vaccinated if the vaccine was made available. Non-acceptance of the COVID-19 vaccine was associated with poor knowledge in our study, similar to the findings in other countries [89,90]. Of interest is that pupils who would accept the COVID-19 vaccine had good knowledge regarding the vaccine compared to those who were hesitant (68.5% vs. 56.3%), similar to studies in Canada, China (including Hong Kong), and Sweden [89,90,91,92]. In addition, in our study, pupils who would accept the COVID-19 vaccine had good attitudes towards the vaccine compared to those who were hesitant (79.1% vs. 33.7%), which is encouraging. Alongside this, participants who were in Grade 9 and had higher scores of knowledge and attitudes, and those who knew a friend or relative who died from COVID-19, also had higher odds of accepting the vaccine.

These findings are consistent with the findings in a systematic review and meta-analysis of studies in sub-Saharan Africa where hesitancy was associated with only attending secondary schools and not higher education [87]. However, this was different to studies in Korea among secondary school pupils (69.1% acceptance) with pupils perceiving the vaccines as safe and effective [93], in China (60% acceptance) [89], and England where more than half (50.1%) of those surveyed were willing to be vaccinated and only 12.9% were hesitant [76]. These appreciable differences between countries could potentially be due to differences in culture, socio-economic status, and robust vaccine promotion messages. In addition, we have seen appreciable vaccine hesitancy among adults across sub-Saharan Africa [39].

Most of the participants in our study accessed information about COVID-19 vaccines through mass and social media, which can be a concern due to the extent of unverified messages [94]. These findings corroborate reports from others indicating the importance of social and mass media in disseminating information and misinformation concerning vaccines [56,95,96,97,98], similar to the situation regarding treatments for patients with COVID-19 [94,98,99]. Social media can potentially be used to increase the awareness of individuals regarding COVID-19 vaccines and change their behaviour [98]. This makes social media platforms potentially one of the best and most efficient platforms for addressing vaccine hesitancy by increasing confidence in vaccine safety and effectiveness. However, any youth-friendly COVID-19 messaging should use pertinent platforms and contain appropriate language style to effectively convey key messages regarding the safety and effectiveness of COVID-19 vaccines [88,100]. Our findings also indicate that the participants accessed information regarding COVID-19 vaccines from HCWs. This is similar to other findings that have reported HCWs, including school nurses, as one of the main sources of reliable and trusted information concerning COVID-19 vaccines [101,102]. This shows that HCWs must champion the promotion of vaccine acceptance and uptake by providing COVID-19 vaccine education, which has not always been the case [46,49,103,104,105]. Given this, steps need to be taken to ensure that HCWs do not enhance hesitancy rates given concerns in some studies, including among African countries [46,47,48,49].

Interestingly in our study, pupils who knew a friend or relative who died from COVID-19 had higher odds of accepting to be vaccinated, which corroborate observations from Pakistan and Italy in which adult participants whose friends or family died due to the COVID-19 pandemic had higher odds of accepting the vaccine [106,107]. Such observations can potentially be used in future messaging campaigns to pupils and their parents. However, in other studies, adults who lost a loved one due to the COVID-19 pandemic did not typically see a need to be vaccinated [108]. In contrast, adolescents in one study who strongly believed that COVID-19 is a high-risk infection and can lead to death, had higher vaccine acceptance rates [93].

The authors are aware of some limitations of this study. Firstly, it was only conducted in Lusaka, which may affect the generalisation of findings to the rest of the secondary schools in the country. Secondly, this was a survey rather than an in-depth discussion with pupils. However, despite this, we believe the findings are robust, providing direction for future nationwide studies.

## 5. Conclusions

This study found a low COVID-19 vaccine acceptance among secondary school children and adolescents in Lusaka City, Zambia. Despite most of the pupils having good scores for knowledge and attitudes, and all of them had heard about COVID-19 vaccines, their low acceptance of the vaccine is of public health concern. The current findings demonstrate the need for heightened vaccine uptake campaigns in secondary schools throughout Zambia, which has started to be enacted. We will be following this up in future studies.

## Figures and Tables

**Table 1 vaccines-10-02141-t001:** Socio-demographics, knowledge and attitude scores by vaccine acceptance.

Variables	Total Population (N = 998); n (%)	COVID-19 Vaccine Acceptors; n (%)	
		No (n = 871)	Yes (n = 127)
Sex			
Female	646 (64.7)	561 (64.4)	85 (66.9)
Male	352 (35.3)	310 (35.6)	42 (33.1)
Living with			
Guardian	43 (4.3)	37 (4.3)	6 (4.7)
Parents	955 (95.7)	834 (95.8)	121 (95.3)
School level			
Grade 8	185 (18.5)	146 (16.8)	39 (30.7)
Grade 9	200 (20.0)	182 (20.9)	18 (14.2)
Grade 10	163 (16.3)	145 (16.7)	18 (14.2)
Grade 11	252 (25.3)	219 (25.4)	33 (26.0)
Grade 12	198 (19.8)	179 (20.6)	19 (15.0)
Source of information about COVID-19			
Healthcare workers			
No	651 (65.2)	575 (66.0)	76 (59.8)
Yes	347 (34.8)	296 (34.0)	51 (40.2)
Mass media (TV/radio)			
No	469 (47.0)	407 (46.7)	62 (48.8)
Yes	529 (53.0)	464 (53.3)	65 (51.2)
Social media			
No	635 (63.6)	546 (62.7)	89 (70.1)
Yes	363 (36.4)	325 (37.3)	38 (30.0)
Family/friends			
No	749 (75.0)	652 (75.0)	97 (76.4)
Yes	249 (25.0)	219 (25.0)	30 (23.6)
Age			
Mean [95% CI]	15.6 [15.5–15.7]	15.7 [15.6–15.8]	15.3 [15.0–15.6]
Total knowledge score %			
Mean [95% CI]	57.8 [56.1–59.5]	56.3 [54.5–58.0]	68.5 [63.5–73.5]
Total attitude score %			
Mean [95% CI]	39.4 [37.4–41.5]	33.7 [31.6–35.7]	79.1 [74.6–83.5]

**Table 2 vaccines-10-02141-t002:** COVID-19 experiences of respondents according to vaccine acceptance.

Experiences/Condition	Total Population (N = 998); n (%)	COVID-19 Vaccine Acceptors; n (%)	
		No (n = 871)	Yes (n = 127)
Suffered from COVID-19 before			
I do not know	45 (4.5)	41 (4.7)	4 (3.2)
No	855 (85.7)	747 (85.8)	108 (85.0)
Yes	98 (9.82)	83 (9.5)	15 (11.8)
Friend/relative suffered from COVID-19			
I do not know	60 (6.0)	60 (6.9)	0
No	471 (47.2)	420 (48.2)	51 (40.2)
Yes	467 (46.8)	391 (44.9)	76 (59.8)
A friend/relative died from COVID-19			
I do not know	107 (10.7)	99 (11.4)	8 (6.3)
No	733 (73.5)	644 (74.0)	89 (70.0)
Yes	158 (15.8)	128 (14.7)	30 (23.6)
Quarantined as a result of COVID-19			
I do not know	95 (9.5)	87 (10.0)	8 (6.3)
No	710 (71.1)	624 (71.6)	86 (67.7)
Yes	193 (19.3)	160 (18.4)	33 (26.0)
Able to practice physical and social distancing			
I do not know	52 (5.2)	49 (5.6)	3 (2.4)
No	167 (16.7)	151 (17.3)	16 (12.6)
Yes	779 (78.1)	671 (77.0)	108 (85.0)
Preventive measures were stressful to follow			
I do not know	85 (8.5)	76 (8.7)	9 (7.1)
No	558 (55.9)	481 (55.2)	77 (60.6)
Yes	355 (35.6)	314 (36.1)	41 (32.3)
Suffer from a chronic condition			
I do not know	52 (5.2)	50 (5.7)	2 (1.6)
No	888 (89.0)	776 (89.1)	112 (88.2)
Yes	58 (5.8)	45 (5.2)	13 (10.2)

**Table 3 vaccines-10-02141-t003:** Association between respondent’s knowledge and attitude scores and acceptance of COVID-19 vaccines, adjusting for potential confounding variables.

Characteristics	OR Adjusted for Age [95% CI]	*p*-Value	Adjusted OR [95% CI]	*p*-Value
Knowledge score	5.44 [2.41–12.3]	<0.001	11.75 [6.51–21.2]	0.001
Attitude score	11.19 [4.87–24.82]	<0.001	9.85 [4.35–22.2]	<0.001
Socio-demographics				
Age	0.90 [0.82–1.00]	0.040	0.85 [0.71–1.01]	0.072
Sex				
Female	Ref			
Male	0.94 [0.64–1.37]	0.730		
Living with				
Guardian	Ref			
Parents	0.80 [0.33–1.89]	0.604		
School level				
Grade 8	Ref		Ref	
Grade 9	0.38 [0.21–0.70]	0.002	0.45 [0.22–0.93]	0.031
Grade 10	0.51 [0.26–1.00]	0.049	0.60 [0.26–1.40]	0.235
Grade 11	0.65 [0.33–1.29]	0.217	0.86 [0.36–2.09]	0.743
Grade 12	0.47 [0.21–1.05]	0.067	0.77 [0.29–2.07]	0.610
Source of information about COVID-19				
Healthcare workers				
No	Ref			
Yes	1.28 [0.89–1.86]	0.889		
Mass media (TV/radio)				
No	Ref			
Yes	0.91 [0.63–1.31]	0.605		
Social media				
No	Ref			
Yes	0.72 [0.49–1.08]	0.114		
Family/friends				
No	Ref			
Yes	0.86 [0.56–1.34]	0.516		
COVID-19 experiences				
Suffered from COVID-19 before				
I do not know	Ref			
No	1.47 [0.53–4.10]	0.464		
No	1.84 [0.59–5.78]	0.294		
A friend/relative died from COVID-19				
I do not know	Ref		Ref	
No	1.76 [0.84–3.67]	0.132	2.53 [0.93–7.34]	0.172
No	2.85 [1.28–6.37]	0.010	3.27 [2.14–5.09]	0.013
Quarantined as a result of COVID-19				
I do not know	Ref			
No	1.54 [0.74–3.21]	0.252	1.42 [0.64–3.19]	0.390
Yes	2.36 [1.07–5.20]	0.033	1.72 [0.69–4.25]	0.243
Able to practice physical and social distancing				
I do not know	Ref			
No	1.81 [0.52–6.32]	0.352	2.44 [0.48–12.3]	0.280
Yes	2.29 [0.87–8.95]	0.084	5.28 [0.90–8.50]	0.065
Preventive measures are stressful to follow				
I do not know	Ref			
No	1.42 [0.69–2.91]	0.337		
Yes	1.14 [0.54–2.41]	0.720		
Suffer from a chronic condition				
I do not know	Ref			
No	3.73 [0.94–14.84]	0.062		
Yes	7.37 [1.66–32.78]	0.009		

NB: 95% CI—95% confidence interval, OR—odds ratio.

## Data Availability

Further data is available on reasonable request from the corresponding author.

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
