# Peer review of "Knowledge, Attitudes, and Acceptance of COVID-19 Vaccines among Secondary School Pupils in Zambia: Implications for Future Educational and Sensitisation Programmes"

_vaccines, 2022, doi:10.3390/vaccines10122141_

Round 1
Reviewer 1 Report
The article is of good quality and clear. I recommend this paper to be published in the journal. Here are some minor suggestions:
1: It is suggested to add some background of this study in introduction and highlight the novelty of this work clearly. Please added relevant statements: “Effective measures are greatly needed to reduce human-to-human transmission. Although in vitro studies (Nat. Microbiol. 2022, 7, 716-725; Eur. J. Med. Chem. 2022, 240, 114570; Front. Pharmacol. 2022, 13, 926507; Sci. Transl. Med. 2022, 14, eabm3410.) have reported some compounds against SARS-CoV-2, their efficacy and safety remain to be further confirmed. At present, promising magic bullets still do not exist (Nature. 2020, 586, 113-119; J. Med. Virol. 2022, 94, 1766-1767; Nat. Rev. Drug Discov. 2022, 21, 3-5.). As an indispensable therapeutic strategy, vaccines have attracted significant attention in countering SARS-CoV-2 infection.” This is critical to address in this manuscript, the authors should enrich this part in the revised version.
2: “Mutations (Alpha, Beta, Gamma, Delta, and Omicron variants) are challenging the efficacy and safety of vaccines.” The authors should emphasize the relevant content.
3: In page 9 line 248-250, “Canada, China, Hong Kong and Sweden” Please correct to “Canada, China (including Hong Kong), and Sweden”
Author Response
Dear reviewer,
Thank you for the comments, find attached our responses.

Reviewer 2 Report
Overall this is a valuable study. It is unfortunate that the authors did not include data on the % of students who were vaccinated. This likely could still be added since student who were vaccinated were excluded and therefore must be known to the researchers.
If doing this study over, it would have been nice to compare the knowledge and attitudes of those who were vaccinated with this group. Also would have been nice to have another vaccine to compare to the COVID vaccine among these students.
It would be nice to have the survey questions in the manuscript text, at least in general so that we know what was included in knowledge and attitude domains.
General issues---the introduction is too long and could be shortened if the information related to international issues, African issues and Zambia issues were combined instead of being presented in series.
Please check the dates for the survey to make sure they are the same in the abstract and the Methods.
Line 194---really not clear whether or not COVID vaccine was readily available to these students. Would be nice to include in the introduction.
Line 211--very minor but you use preventative sometimes and preventive others. The preferred term is preventive.
Line 286-288--where is data that says HCW were vital sources of information?
Author Response
Dear reviewer,
Thank you for your review and comments, the comments helped us improve our paper. Please find attached our responses.
